# Depression, Anxiety and Stress on Caregivers of Persons with Dementia (CGPWD) in Hong Kong amid COVID-19 Pandemic

**DOI:** 10.3390/ijerph19010184

**Published:** 2021-12-24

**Authors:** Tommy Kwan-Hin Fong, Teris Cheung, Wai-Chi Chan, Calvin Pak-Wing Cheng

**Affiliations:** 1Department of Psychiatry, The University of Hong Kong, Hong Kong, China; tommykhf@hku.hk (T.K.-H.F.); waicchan@hku.hk (W.-C.C.); 2School of Nursing, The Hong Kong Polytechnic University, Hong Kong, China; teris.cheung@polyu.edu.hk

**Keywords:** COVID-19, CGPWD, mental health, depression, anxiety, stress

## Abstract

Background: Coronavirus disease 2019 (COVID-19) contributed to increasing prevalence of depressive symptoms and other psychological repercussions, particularly in the disease population in Hong Kong. Nonetheless, the caregiver burden of caregivers of persons with dementia (CGPWD), has been under-investigated. Aims: This study examined the psychological impact and its correlates on the CGPWD in Hong Kong amid the COVID-19 outbreak. Methods: CGPWD referred from rehabilitation clinics and online seminar were used to recruit participants to complete an online questionnaire by the end of the second-wave of the COVID-19 outbreak (June 2021). To be eligible, either full-time or part-time CGPWD, aged 18 or above, can understand Cantonese, currently reside in Hong Kong and offering care to PWD for at least one year, were recruited. Those CGPWD diagnosed with any type of psychiatric disorder were excluded from this study. The Chinese Center for Epidemiologic Studies Depression Scale (CES-D), Perceived Stress Scale (PSS-10), Generalized Anxiety Disorder (GAD-7), Zarit Burden Interview (ZBI-22), and Nonattachment Scale (NAS-7) were used to measure participants’ depression, perceived stress, anxiety symptoms, caregiver burden and wisdom in subjective feelings of internal stress. The modified Medical Outcomes Study Social Support Survey (mMOS-SS) and the SARS Appraisal Inventory (SAI) were also administered to measure participants’ perceived support and coping efficacy. Follow-up responses were gathered by the end of third-wave outbreak (October 2021). Results: A total of 51 CGPWD participated, of which, 33 (64.7%) suffered from probable depression (CES-D score ≥ 16). Participants also showed a significant increase in depression symptom scores at the three-month follow-up period (t = 2.25, *p* = 0.03). CGPWD with probable depression had less non-attachment awareness and higher scores in anxiety, stress, caregiving burden, and coronavirus impact (all *p* < 0.05) than those without. Conclusions: High prevalence of depressive symptoms was noted among our CGPWD sample and these symptoms seemed to worsen substantially. Contingent online mental health support should be prioritized to those CGPWD to reduce psychiatric morbidity and the global disease burden.

## 1. Introduction

The coronavirus disease 2019 (COVID-19) has been rapidly transmitted to more than 220 countries nationwide. Hong Kong is no exception. As of 18 December 2021, Hong Kong has had 12,513 total COVID-19 cases, with 213 deaths [1]. Comparing the total number of COVID-19 cases of more than 27 million nationwide with 5.33 million deaths, the situation in Hong Kong represents but the tip of the iceberg [2].

The rapid transmission of this novel virus has inevitably created an apprehensive atmosphere, widespread anxiety and uncertainty to the general public [3]. Citizens are confronted with this highly contagious novel virus which may bring about irreversible damage, jeopardizing both physical and mental health [4]. The enforcement of stringent infection control measures in affected countries/regions including compulsory face mask wearing, social distancing, quarantines, lockdowns, and closure of entertainment facilities and sports amenities may lead to social withdrawal, anxiety and fear, boredom, loneliness, anger [3] and depression [5].

Notwithstanding the socio-economic burden brought by the COVID-19 pandemic, it also adds enormous physical and psychological burden on existing healthcare systems, particularly on health professionals, infected patients, suspected cases and their family caregivers [6]. The general population is also affected, especially during the lockdown periods; non-emergency community or day service centres were temporarily suspended in Hong Kong. Suspension of existing services will add more strain on caregivers during the pandemic. Prolonged physical exhaustion may jeopardize family caregivers’ mental well-being. Emerging local evidence from a recent cross-sectional survey [7] found that the anxiety levels of Hong Kong citizens are hitting record highs, with 88% of respondents (N = 1168) reporting borderline abnormal anxiety (88%) when they have possible contact with COVID-19 cases. Another large-scale cross-sectional study which surveyed 11,072 Hong Kong adults from March 2020 to April 2020 discovered a high point-prevalence of probable depression and suicidal traits during the COVID-19 outbreak, and those wearing face masks for self-protection were more prone to depression [8]. Most COVID-19 infections are not severe [9]. In fact, the majority of the fatal cases usually occurred in elderly patients with underlying medical comorbidities [10]. Thus, it is understandable that older adults may have more fear of infection as they have higher risks of susceptibility to COVID-19 and a higher chance of it developing into severe medical condition [11]. In fact, one of the most prevailing health conditions associated with aging is dementia. Conceivably, the outbreak of COVID-19 will afflict persons with dementia and their caregivers by restricting daily living as well as aggravating their psychological concerns. 

Notably, a significant proportion of older adults in Hong Kong are suffering from dementia. It is estimated that every 5 to 8 adults aged 65 or above, suffer from dementia in every 100 older adults, and the ratio will spiral up to every 1 in 5 per 100 older adults if they aged over 80 years old [12]. More importantly, caregivers of persons with dementia (CGPWD) are predominantly middle aged or older age adults themselves. Thus, it is unsurprising that CGPWD are more vulnerable to depression or anxiety [13]. The higher the dependency level of the dementia patient in activities of daily living and the more severe and frequent behavioural symptoms, the more likely the CGPWD will develop depressive symptoms [14]. Caregiving accounted for high levels of stress among CGPWD in Hong Kong [15]. There is ample empirical evidence supporting the negative detrimental impact of the caregiving burden; however, the psychological impact on those CGPWD in Hong Kong during the COVID-19 pandemic has not been thoroughly investigated. 

Despite some evidence demonstrating the negative detrimental impact of COVID-19 on the caregiving burden, research evidence is insufficient to influence policymaking. For instance, an online cross-sectional study in Italy involving 84 CGPWD discovered higher caregiving burden, depression and anxiety levels during the COVID-19 lockdown period [16]. Another web survey conducted in Japan on 635 CGPWD investigating their quality of life, caregiving burden and work productivity. Results showed that 50.5% of the primary caregivers had significantly lower quality of life and higher work impairment than non-primary caregivers [17] while another local qualitative study shared similar findings regarding caregiving burden on stroke caregivers [18]. Nonetheless, there is by far no local study investigating the impact of COVID-19 in the long-term caregiving on CGPWD in Hong Kong.

Since late March 2020, quarantine measures and social distancing have been imposed by the Hong Kong government, such as restrictions on social gatherings and temporarily closure of recreational facilities and sports amenities [19]; these stringent measures are still in force at the time of reporting. With prolonged caregiving duration, CGPWD caregivers would have limited time for relaxation, and accumulated stress and anxiety may trigger depressive symptoms if they go untreated in a timely fashion. A high level of caregiving burden would also contribute to caregivers’ poor self-rated health, adverse health behaviours, and increased visits to healthcare services [20]. Such claims have been supported by a nationwide survey involving 4913 dementia caregivers in Italy, with more than half of respondents showing exacerbated behavioural and psychological symptoms after one month of quarantine [21]. Conceivably, we could postulate that the prolonged COVID-19 situation aggravated depressive symptoms of CGPWD [8]. Therefore, it is imminent to examine the psychological impacts of COVID-19 on CGPWD and investigate the conceivable elements that may help planning early mental health prevention and intervention amid the COVID-19 pandemic in Hong Kong. The aim of this study was three-fold: first, to determine whether the mental health of CGPWD in Hong Kong was adversely influenced by the second-wave of the COVID-19 outbreak (March 2020 to May 2020); second, to examine the psychological status of these CGPWD by the end of the third-wave of the COVID-19 outbreak (July 2020 to September 2020); and third, to identify the prevalence and correlates of outcome variables (i.e., depression, perceived stress, anxiety, perceived social support, coping efficacy) among the CGPWD.

## 2. Methods

### 2.1. Study Design

This was a cross-sectional study using convenience sampling method. Participants were instructed to complete a baseline questionnaire in traditional Chinese, the official written language in Hong Kong. Follow-up responses were collected by the end of the third-wave of the COVID-19 outbreak (July 2020 to September 2020).

### 2.2. Subjects

To be eligible, participants should be: (1) able to read/understand Cantonese (official spoken language in Hong Kong) and currently reside in Hong Kong; (2)aged 18 or above; (3) full-time or part-time informal caregivers (e.g., unpaid family members) who offer care towards the PWD for one year or more. CGPWD diagnosed with any types of psychiatric illnesses or currently receiving psychotherapy were excluded from this study. 

### 2.3. Ethical Considerations

This study was approved by the Institutional Review Board of The University of Hong Kong/Hospital Authority, Hong Kong West Cluster (HKU/HA HKW IRB) (reference no.: UW 20-377). All participants provided their written informed consent prior to participating in this study. Participants were assured of their confidentiality, anonymity, and rights of withdrawal.

### 2.4. Data Collection

Recruitment was started using QR code posters flagged up in the David Trench Rehabilitation Centre in the Queen Mary Hospital, Hospital Authority, Hong Kong since early June 2021. Eligible participants were directed to fill in an online screening questionnaire. The data collection process was obscured due to the third COVID-19 outbreak in July 2021. Hence, we recruited potential participants via an online seminar targeting CGPWD organized in early August 2021. Given the fact that the third wave of COVID-19 outbreak was contained in late September 2021, eventually a follow-up questionnaire was delivered to participants in October 2021 using either telephone interviews or online surveys. The data collection period spanned from June 2021 and October 2021.

### 2.5. Outcome Measures

#### 2.5.1. Demographics

Subjects’ demographics, including gender, age, marital status, educational level, living circumstances, occupation, and monthly personal income, financial status, personal and family history of mood disorders, relationship with PWD, and caring durations were canvassed. 

#### 2.5.2. COVID-19 Related Attitudes

Due to the lack of validated instruments to solicit CGPWD’s perceptions towards COVID-19 patients, we thus developed a new eight-item rating scale, jointly designed by a team of psychiatrists who are experts in gerontology in the Department of Psychiatry, University of Hong Kong. This scale scored from 1 (absolutely disagree) to 9 (absolutely agree). Participants were asked to rate statements like “I am so scared of persons with coronavirus” and “People infected with coronavirus are disgusting”. Scores ranged from 9 to 72. Higher scores indicate more negative attitudes towards COVID-19 patients.

#### 2.5.3. Depressive Symptoms

Depressive symptoms were examined by a validated self-report 20-item questionnaire, the Chinese Center for Epidemiologic Studies Depression Scale (CES-D) [22,23]. Scores ranged from 0 to 60, with higher scores indicating more severe depressive symptoms. Participants with a total CES-D score of ≥16 were categorized as having probable depression. The scale showed satisfactory construct validity in the Chinese population [23].

#### 2.5.4. Perceived Stress

Perceived stress levels were measured by a 10-item Perceived Stress Scale (PSS-10) [24]. The PSS-10 used a 5-point Likert scale ranging from 0 (never) to 4 (very often), with a higher score indicating a higher stress level. This scale has satisfactory psychometric properties and good reliability [25]. 

#### 2.5.5. Anxiety

The level of anxiety was assessed by a self-rated seven-item Generalized Anxiety Disorder (GAD-7) [26]. GAD-7 uses a 4-point Likert scale ranged from 0 (not at all sure) to 4 (nearly every day). A higher score indicates a higher anxiety level. This scale is proven to have a good reliability and validity in measuring anxiety among the Chinese population [27].

#### 2.5.6. Caregiver Burden

Levels of caregiver burden were measured by a self-administrated Zarit Burden Interview—ZBI-22 [28]. Score ranges from 0 to 88, with higher scores indicating higher degree of caregiving burden. The Chinese version of the scale demonstrated good validity and reliability among the CPGWD in Hong Kong [29].

#### 2.5.7. Degrees of Non-Attachment Traits

The seven-item Nonattachment Scale (NAS-7) was used to assess participants’ levels of wisdom in subjective detachment feelings on internal pressure or experiences. We used the Chinese version of the NAS-7, as it has excellent internal consistency in previous studies [30]. A higher score indicates a higher degree of nonattachment. 

#### 2.5.8. Perceived Social Support

Perceived social support was measured by the modified Medical Outcomes Study Social Support Survey (mMOS-SS), which comprised eight questions covering emotional and instrumental social support domains. The internal reliability and consistency, construct and discriminant validity were satisfactory [31]. Higher scores indicated more social support [32]. The Chinese version of the mMOS-SS has good psychometric properties [33].

#### 2.5.9. Perceived Coping Efficacy

Perceived coping efficacy was assessed by a COVID-19 coping efficacy inventory (2019-nCoV-CEI) modified from the SARS Appraisal Inventory (SAI). This scale assessed eight daily life aspects. Participants were asked to rate their level of confidence in dealing with the pandemic influence on a 5-point scale (0: none, 4: very high). The internal consistency of the SAI was satisfactory [34].

### 2.6. Statistical Analysis

Statistical analyses were performed using IBM SPSS Statistics Version 26.0 for the Windows platform (SPSS Inc., Chicago, IL, USA). The normality of the data distributions were assessed by QQ plot. Descriptive analysis was used to describe participants’ sociodemographic characteristics. Between-group independent samples *t*-tests were used to compare the scores of psychological instruments (CES-D, PSS-10, GAD-7, ZBI-22, NAS-7, mMOS-SS and SAI). Level of significance was set at *p* < 0.05 (two-tailed). Participants were categorized into two groups (Depression and non-depression group) using the cut-off point of CESD ≥ 16. Pearson’s correlation analysis and backward multiple regression models were carried out to investigate potential confounding factors on CGPWD’s instrumental ratings. Paired samples *t*-tests were used to compare the pre-and-post scores on depressive symptoms, perceived stress, anxiety level, and perceived coping efficacy in different pandemic waves. Missing data was handled by multiple imputation [35].

## 3. Results

A total of 51 CGPWD caregivers participated in this study. Since only completed responses were recorded online, we cannot estimate the overall response rate. Table 1 described the sociodemographic characteristics and baseline instrumental scores of the entire samples across the depression and non-depression group. Participants were predominantly female (n = 44, 86.3%). Participant ages varied from 25 to 92 years of age. The mean age was 53.54. About half of the sample were single (49.0%), of which 56.9% obtained a Bachelor degree or above. Around two-thirds (62.7%) lived in private housing. A vast majority of participants (88.2%) either lived alone or living with Persons with Dementia (PWD). Around 40% (41.2%) had a monthly personal income of less than HKD 20,000, with 43.1% having a full-time job. One-fifth (21.6%) had a personal or family history of mood disorders. More than three-quarters of respondents (76.4%) reported that their parents suffered from dementia. Duration of caring for the PWD seemed dispersed in our sample, with most caregivers (45.1%) having less than 10 years of (i.e., 5 to 9 years) of caring experience. More than half of the participants (54.9%) received community service assistance (e.g., caregiving assistance from non-governmental organisations), while another one-third (33.3%) sought informal help from friends/ relatives. 

Regarding the point prevalence of depressive symptoms for CGPWD, participants’ mean total CES-D score was 19.76 (*SD* = 11.69), suggesting the likelihood of having moderate depression. Nearly two-third of respondents (n = 33, 64.7%) were identified with high risks of probable depression (total CES-D score ≥16), and there was a statistically significant difference in mean score between the depression and the non-depression group (*t* = 7.55, *p* < 0.001). Results from independent samples *t*-tests also showed significant differences on perceived stress, anxiety, caregiving burden, non-attachment, and perceived pandemic impact between depression and non-depression groups (all *p* < 0.05). Chi-square analyses revealed statistically significant correlations between monthly income (*p* < 0.001) and personal history of mood disorders (*p* = 0.006).

Table 2 showed the results of Pearson correlational matrix across the psychometric variables. Our findings revealed that caregivers’ attitudes on COVID-19 was significantly related to depressive symptoms, perceived stress, anxiety, caregiving burden, and confidence in pandemic coping (all *p* < 0.05). CGPWD who perceived the pandemic as more severe were associated with higher levels of depressive symptoms, perceived stress, anxiety, and caregiving burden. In order to identify the parsimonious combination of variables contributing towards the psychological distress, backward multiple regression analyses were conducted by using the total scores of CES-D, PSS-10 and GAD-7 as dependent variables. The ratings on ZBI-22, NAS-7, mMOS-SS, COVID-19 attitudes, and perceived impact were entered in the regression model as predictors. The regression analyses are summarized in Table 3.

Overall, the final models regarding depression, stress, and anxiety account for 58.7%, 41.5% and 55.1% of variance, respectively. The final model had identified caregiving burden, non-attachment, and attitudes on COVID-19, with caregiving burden and non-attachment as significant predictors of depression (*F* (3,47) = 22.24, *p* < 0.001). The stress model with the most parsimonious predictor variables included caregiving burden, attitudes on COVID-19, and perceived impact, with caregiving burden identified as significant contributor (*F* (3,47) = 11.11, *p* < 0.001). For anxiety, caregiving burden was the only significant predictor (*F* (1,49) = 60.14, *p* < 0.001).

For the follow-up survey, 43 responses were obtained. Table 4 presented the results of paired sample *t*-tests on depression, stress, anxiety, and perceived coping efficacy. Participants demonstrated significantly higher depressive symptoms as evaluated by CES-D after the third wave outbreak (*t* = 2.25, *p* = 0.03). An increase in perceived stress (mean difference = 0.42) and anxiety (mean difference = 0.37) was also noted despite the fact that these figures were statistically insignificant. The trend of having lessening concerns in epidemic impact was observed (from 1.83 to 1.69), yet the overall confidence in coping efficacy decreased (from 2.41 to 2.28).

## 4. Discussion

It is evident that our CGPWD manifested moderate depressive symptoms in caring the PWD during the second and third wave amid COVID-19 outbreak in Hong Kong. The prevalence rate of probable depression among the 51 dementia caregivers was 64.7% in this study, which is even higher than another cross-sectional study conducted in the general population in Hong Kong in an earlier COVID-19 outbreak period in Hong Kong, which revealed that 46.5% (n = 11, 072) of participants suffered from probable depression (using a cut off PHQ-9 score ≥10) [8]. The mean total of CES-D and ZBI-22 scores in our study are also higher than the scores that emanated from an American study beyond the pandemic period (CES-D total score: 19.76 vs 16.09; ZBI-22 total score: 40.78 vs 38.61) [36], suggesting a remarkable deterioration of mental well-being precipitated by the COVID-19 outbreak. An increase in depressive symptoms among CGPWD is also noted in the follow-up survey, which is not anticipated considering the cumulation of the third coronavirus outbreak. In addition, the results consolidate the verdict on exacerbated behavioural and psychological symptoms during COVID-19 quarantine circumstance despite contextual differences [20].

CGPWD with probable depression in this study also report high levels of anxiety, perceived stress, and caregiving burden, along with a more pessimistic attribute on non-attachment idiosyncrasy impacted by the coronavirus. Results obtained from this study further affirm that the caregiving burden and anxiety of the CGPWD should not be under-estimated, as our results echoed another Chinese study conducted on family caregivers of schizophrenia individuals [35] which revealed the mean score of 48 using the ZBI-22. This score was very close to our ZBI-22 score among the depression group (*M* = 47.9). The mean GAD-7 score of Yu’s study was 9.3 (*SD* = 6.6), which is also analogous to our GAD-7 score (*M* = 9.2, *SD* = 4.1) despite the disparity in the caregiving nature [37].

The high prevalence of depressive symptoms reported from CGPWD in this study delivered a significant message to policymakers and healthcare providers. The emergence of COVID-19 has a high resemblance to the outbreak of the 2003 severe acute respiratory syndrome (SARS) in Hong Kong, which claimed 299 lives [38]. It is unsurprising that the general population in Hong Kong strictly complied with the stringent infection control measures, especially those who had experienced the SARS outbreak, for instance, with the wearing of face masks and frequent handwashing and compulsory quarantines etc. These imminent infection control measures have inevitably re-triggered their past fear towards the 2003 SARS outbreak that subsequently intensified Hong Kong citizens’ psychological distress and anxiety [6]. The rapid transmission of this novel virus has triggered intense fear, stress, anxiety and depression, particularly if individuals perceive themselves as more susceptible to COVID-19 infection than others [39]. Negative news portrayal and insufficient knowledge on the transmission route and precautionary measures towards COVID-19 may trigger further uncertainty and negative psychological symptoms [40]. The relatively high level of depressive symptom score implies that CGPWD with high caregiving burden and less non-attachment awareness have increased likelihood of reporting more severe depressive symptoms. The lack of non-attachment feeling on internal pressure and adversities by caregivers could intensify their sentiment on the impact of COVID-19 and caregiving burden in manipulating potential psychological distress. The stringent quarantine measures might lead to increased caregiving duration, resulting in deprived personal private time for CGPWD. This prolonged caregiving burden may increase perceived powerlessness and physical and mental exhaustion that eventually jeopardize CGPWD’s mental wellbeing [41].

Furthermore, participants with probable depression were significantly and inversely associated with monthly income and personal history of mood disorders. Higher depressive symptoms are predominantly present in caregivers with lower monthly income. This phenomenon may partially explain by the socioeconomic downturn brought by COVID-19 in Hong Kong. It is noteworthy that most participants with probable depression did not have a psychiatric history of mood disorders. We speculate that these CGPWD may have poor awareness and mental health literacy of their increasing severity of their psychiatric symptoms and thus, they did not seek professional help in a timely fashion. Poor mental health literacy might lead to undesirable tragedy with possible suicidal ideations or harmful actions to the PWD [42]. Our results indicate that contingent mental health services and intervention including telemedicine and digital online psychiatric consultation [43] by mental health experts should be urgently established (e.g., Zoom consultation and emergency community outreach) to assist the CGPWD to cope with insurmountable psychological pressure amid the COVID-19 pandemic [44]. Delivery of digital/online mental health information or services regarding professional mental health seeking methods, management of crisis interventions, and an emergency 24-h helpline directory for at-risk individuals, should be extensively disseminated in online social media via joint effort by the Department of Health, the Centre for Health Protection, and the Hospital Authority in Hong Kong. These health authorities should form a collaborative alliance to formulate effective, tailor-made interventions specifically for improving CGPWD’s mental health literacy and to enhance professional help-seeking behaviour during the pandemic era.

## 5. Limitations

There are a few limitations that we must acknowledge in this study. Since face-to-face interviews were all suspended during the initial COVID-19 outbreak, we could only rely on online surveys which is not uncommon to generate relatively low enrolment and response rate. Second, the use of QR code enrolment in our online survey may filter out some potential participants, for example, CGPWD with severe depression, and those with technical phobia. Third, prolonged data collection periods limit the accuracy and reliability of the findings given the everchanging coronavirus upheavals in Hong Kong. Fourth, the study did not have any pre-pandemic data on the CGPWD population which may serve as a comparison cohort for this study. Hence, results emerged from a small sample size in this study may not be generalized to other countries. Fifth, due to cross-sectional design, causality between CGPWD and psychological impact cannot be inferred. Despite all these limitations, our findings highlight an important message that CGPWD is also an at-risk group which seems to be under-investigated in the existing literature. More research in this subpopulation is warranted. Future replication of further studies using longitudinal or qualitative design with larger sample size using representative samples is recommended. Despite its relatively small sample size, our study findings have shed insight on mental health professionals and public health policymakers that CGPWD caregivers with probable depression tend to perceive higher anxiety, stress, and caregiving burden alongside with a more pessimistic perspective on COVID-19.

## 6. Conclusions

It is evident that CGPWD have a high prevalence of depressive symptoms as a result of the COVID-19 outbreak in Hong Kong. It is highly likely that participants’ level of depression may further spiral upwards if there is still no timely effective mental health intervention in place in this pandemic era. Apart from the caregiving burden, CGPWD with non-attachment are also more prone to reporting more depressive symptoms. The high point prevalence of probable depression among CGPWD sends a strong message that there is a pressing need of provision of contingent mental health interventions/practical assistance/financial subsidy from allied mental health professionals and social welfare departments in order to restore CGPWD’s mental wellbeing so as to reduce the increasing risk of psychiatric morbidity and global disease burden brought by this COVID-19 pandemic.

## Figures and Tables

**Table 1 ijerph-19-00184-t001:** Demographic characteristics of participants by depression category (N = 51).

		Entire Sample (N = 51)		Depression (N = 33)		No Depression (N = 18)		χ^2^ (df)	*p*-Value
		N	%	N	%	N	%		
Gender									
	Male	7	13.7	3	16.7	4	12.1	0.20 (1)	0.652
	Female	44	86.3	15	83.3	29	87.9		
Age									
	18–30	2	3.9	2	6.1			4.74 (3)	0.192
	31–45	9	17.6	8	24.2	1	5.6		
	46–60	26	51.0	14	42.4	12	66.7		
	61 or above	14	27.5	9	27.3	5	27.8		
Marital status									
	Single	25	49.0	20	60.6	5	27.8	5.44 (2)	0.066
	Married/In a relationship	23	45.1	12	36.4	11	61.1		
	Divorced/Separated/Widowed	3	5.9	1	3.0	2	11.2		
Education level									
	Elementary or below	3	5.9	1	3.0	2	11.1	2.05 (2)	0.358
	High School/College	19	37.2	14	42.4	5	27.8		
	University and above	29	56.9	18	54.5	11	61.1		
Housing									
	Private	32	62.7	19	57.6	13	72.2	3.72 (3)	0.294
	Home Ownership Scheme	9	17.7	5	15.2	4	22.2		
	Public	7	13.7	6	18.2	1	5.6		
	Others	3	5.9	3	9.1				
Living status									
	Living alone/with PWD	45	88.2	28	84.8	17	94.4	1.03 (1)	0.309
	Living with family/others	6	11.8	5	15.2	1	5.6		
Occupation status									
	Full-time	22	43.1	13	39.4	9	50.0	1.44 (3)	0.696
	Part-time	9	17.7	6	18.2	3	16.7		
	Retired	10	19.6	6	18.2	4	22.2		
	Housewives/Others	10	19.6	8	24.2	2	11.1		
Monthly personal income (HK$)									
	<20,000	21	41.2	15	45.5	6	33.3	17.97 (3)	< 0.001 ***
	20,000–39,999	15	29.4	12	36.4	3	16.7		
	40,000–59,999	7	13.7	6	18.2	1	5.6		
	≧60,000	8	15.7			8	44.4		
Sufficient income for daily needs									
	Yes	39	76.5	23	69.7	16	88.9	2.38 (1)	0.123
	No	12	23.5	10	30.3	2	11.1		
Family history of mood disorders									
	Yes	11	21.6	8	24.2	3	16.7	0.40 (1)	0.530
	No	40	78.4	25	75.8	15	83.3		
Personal history of mood disorders									
	Yes	11	21.6	11	33.3	0	0	7.65 (1)	0.006 **
	No	40	78.4	22	66.7	18	100		
Relationship with PWD									
	Husband/Wife	6	11.8	5	15.2	1	5.6	1.49 (2)	0.476
	Parents	39	76.4	25	75.8	14	77.8		
	Others	6	11.8	3	9.1	3	16.7		
Caring periods									
	Below 5 years	17	33.3	12	36.4	5	27.8	1.09 (2)	0.296
	5–9 years	23	45.1	13	39.4	10	55.6		
	10 years or above	11	21.6	8	24.2	3	16.7		
Community aids (e.g., hiring helpers)									
	Yes	28	54.9	16	48.5	12	66.7	0.17 (1)	0.678
	No	23	45.1	17	51.5	6	33.3		
Informal help (e.g., friends)									
	Yes	17	33.3	11	33.3	6	33.3	0 (1)	1
	No	34	66.7	22	66.7	12	66.7		
**Baseline Assessments**		**M**	**SD**	**M**	**SD**	**M**	**SD**	** *t* **	***p*-Value**
Center for Epidemiologic Studies Depression Scale (CES-D)		19.76	11.69	26.03	9.54	8.28	3.80	7.55	<0.001 ***
Perceived Stress Scale-10 (PSS-10)		21.96	4.76	23.00	4.68	20.06	4.41	2.19	0.033 *
Generalized Anxiety Disorder (GAD-7)		6.86	5.21	9.21	4.08	2.56	4.26	5.49	<0.001 ***
Zarit Burden Scale (ZBI-22)		40.78	17.05	47.94	12.90	27.67	16.13	4.91	<0.001 ***
Non-attachment Scale (NAS-7)		30.33	5.87	28.27	5.54	34.11	4.50	−3.83	<0.001 ***
modified Medical Outcomes Study Social Support Survey (mMOS-SS)		21.75	7.06	21.85	6.29	21.56	8.48	0.14	0.889
	Instrumental	10.12	4.17	10.24	3.41	9.89	5.40	0.29	0.776
	Emotional	11.63	3.72	11.61	3.66	11.67	3.94	−0.06	0.956
Perceived Coping Efficacy—Impact		1.83	0.73	2.03	0.66	1.48	0.74	2.72	0.009 **
Perceived Coping Efficacy—Confidence		2.40	0.58	2.32	0.53	2.55	0.66	−1.33	0.188

Note. * *p* < 0.05; ** *p* < 0.01; *** *p* < 0.001 N = total number of participants; M = mean; SD = standard deviation; PWD = persons with Dementia. Chi-squares (χ^2^)/*t*-tests (*t*) comparing depression/non-depression groups. Depression group is defined by a total CES-D cut-off score of ≥16.

**Table 2 ijerph-19-00184-t002:** Pearson correlational matrix regarding CGPWD’s attitude on COVID-19 and their psycho-social wellbeing (N = 51).

Measures	(1)	(2)	(3)	(4)	(5)	(6)	(7)	(8)	(9)
(1) Attitudes on COVID-19	1.000								
(2) Depression	0.481 ***	1.000							
(3) Perceived Stress	0.385 **	0.574 ***	1.000						
(4) Anxiety	0.376 **	0.780 ***	0.600 ***	1.000					
(5) Caregiving burden	0.444 **	0.708 ***	0.592 ***	0.742 ***	1.000				
(6) Non-attachment	−0.188	−0.429 **	−0.059	−0.326 *	−0.284 *	1.000			
(7) Social support	−0.019	−0.017	0.314 *	0.176	0.250	0.148	1.000		
(8) Perceived impact	0.121	0.357 *	0.458 **	0.445 **	0.485 ***	−0.097	0.251	1.000	
(9) Perceived confidence	−0.291 *	−0.161	−0.120	−0.171	−0.160	0.276	0.222	−0.329 *	1.000

Note. * *p* < 0.05, ** *p* < 0.01, *** *p* < 0.001.

**Table 3 ijerph-19-00184-t003:** Backward multiple regressions analyses identifying significant correlates of Depression, Stress and Anxiety among CGPWD (N = 51).

Variable	Depression			Stress			Anxiety		
	β (95% CI)	*p*	*R* ^2^	β (95% CI)	*p*	*R* ^2^	β (95% CI)	*p*	*R* ^2^
Initial Model			0.607			0.449			0.579
Caregiving burden	0.571 (0.217, 0.565)	<0.001 ***		0.379 (0.022, 0.190)	0.015 *		0.614 (0.107, 0.268)	<0.001 ***	
Non-attachment	−0.205 (−0.811, −0.006)	0.047 *		0.086 (−0.124, 0.264)	0.474		−0.129 (−0.300, 0.071)	0.220	
Social support	−0.144 (−0.156, 0.027)	0.160		0.156 (−0.016, 0.072)	0.200		0.012 (−0.040, 0.044)	0.907	
Attitudes on COVID-19	0.177 (−0.047, 0.500)	0.102		0.209 (−0.023, 0.241)	0.103		0.065 (−0.089, 0.163)	0.559	
Perceived impact	0.075 (−2.297, 4.709)	0.492		0.219 (−0.265, 3.112)	0.096		0.123 (−0.735, 2.495)	0.278	
Final Model			0.587			0.415			0.551
Caregiving burden	0.557 (0.234, 0.530)	<0.001 ***		2.760 (0.030, 0.190)	0.008 **		0.742 (0.168, 0.285)	<0.001 ***	
Non-attachment	−0.236 (−0.861, −0.076)	0.020 *		—	—		—	—	
Attitudes on COVID-19	0.189 (−0.028, 0.513)	0.078		1.440 (−0.037, 0.226)	0.157		—	—	
Perceived impact	—	—		1.915 (−0.081, 3.287)	0.062		—	—	

Note: * *p* < 0.05, ** *p* < 0.01, *** *p* < 0.001, β = regression coefficient; *R*^2^ = R squared; CI = confidence interval.

**Table 4 ijerph-19-00184-t004:** Paired samples *t*-tests comparing baseline and follow-up assessment scores (N = 43).

	Baseline		Follow-Up		*t*	*p*-Value
	M	SD	M	SD		
Center for Epidemiologic Studies Depression Scale (CES-D)	19.40	12.12	22.40	11.10	2.25	0.030 *
Perceived Stress Scale-10 (PSS-10)	22.02	5.11	22.44	3.72	0.59	0.558
Generalized Anxiety Disorder (GAD-7)	6.79	5.24	7.16	4.39	0.53	0.598
Perceived Coping Efficacy—Impact	1.83	0.77	1.69	0.73	−1.31	0.198
Perceived Coping Efficacy—Efficacy	2.41	0.61	2.28	0.71	−1.27	0.211

Note. * *p* < 0.05. M = mean; SD = standard deviation. *t*-tests (*t*) comparing baseline and follow-up data.

## Data Availability

Anonymous data is available on request.

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
