# Peer review of "Depression, Anxiety and Stress on Caregivers of Persons with Dementia (CGPWD) in Hong Kong amid COVID-19 Pandemic"

_ijerph, 2021, doi:10.3390/ijerph19010184_

Round 1
Reviewer 1 Report
Very well done. Minor suggestion- clarify if traditional chinese and cantonese languages are the same or how each were used (methods section). Some font size variations.
Reviewer 2 Report
Dear Authors
I found your paper interesting, thank you. I think caregiver mental health is an important and as you point out, overlooked research area.
Strengths:
-The paper covers an Important topic- especially as carers of people with complex needs such as dementia is not a well-researched area. it is important to put the mental health and coping needs of this group of people on the health agenda.
-The methodology and analysis are adequate and appropriate for the research questions.
I thought the methodology and statistics were sound.
There is a moderate amount of English editing that needs to occur to make the paper easily read. The English grammar is in needs of moderate levels of editing. The phrases and sentences are often not translated correctly, or are missing words or have incorrect temporal tenses. I have pointed out a few instances in the PDF of the paper (attached) where the tense or the grammar are incorrect or inconsistent. In other places, the meaning is obscured by the language used.
Also be careful of the conclusions you make so the results are not over-stated. The conclusion may be over-reaching and needs to be in line with the limitations of the cross-sectional nature of the study.
Thank you and all the best with your research.

Reviewer 3 Report
Hin Fong and collaborators aimed to investigate the psychological impact of the COVID-19 outbreak in Hong Kong caregivers. Although the research topic is relevant, the very limited sample size and unclear novelty of the study do not support the conclusions claimed by the authors. Furthermore, the manuscript requires several edits. Specifically, I have the following recommendations to improve the manuscript:
Overall
The text is difficult to follow due to grammatical mistakes. Please, review the manuscript carefully.
The sample size is very small (also, no power calculation is provided).
Abstract
Please define correctly COVID-19, it is Coronavirus disease 2019 instead of only coronavirus disease
There is a difference between the novel coronavirus (2019-nCoV) and the COVID-19 disease. The 2019-nCoV is the causal agent of the COVID-19 disease. Please consider this difference and do not interchange the terms.
Please define the following abbreviations: CES-D, PSS-10, GAD-7, ZBI-22, NAS-7, mMOSS-SS and SAI before their use in the abstract.
Introduction
Please provide references for the first two sentences of the second paragraph.
Please provide references and make a brief summary of the evidence suggesting that there is an impact of COVID-19 on the caregiver burden.
Please, clarify if there is any study about the impact of COVID-19 on the caregiver burden in HK.
Methods
Provide more details about the COVID-19 related attitudes questionnaire. Has this questionnaire been previously validated? How was it developed? Has it been used in previous studies? What is its reliability?
Has GAD-7 been validated in the Chinese population?
The Pearson correlation analysis is not mentioned in the Methods, only in the results. Please describe this analysis in the Methods section
Please provide a power calculation
Was the normal distribution of the data assessed?
Pearson correlation calculations do not allow the inclusion of relevant covariates to consider in this context such as socioeconomic status. What is the reasoning for its use instead of methods that allow the inclusion of covariates?
How did the authors handle missing data?
Please provide the reasoning to exclude caregivers who have been diagnosed with any psychiatric disorders or currently receiving psychotherapy.
Results
It is mentioned by the authors that “After excluding samples with previously diagnosed mood disorders, the mean differences between depression and non-depression groups on aforementioned psychometric measures apart from perceived stress remained significant”. Is it supposed that caregivers who have been diagnosed with any psychiatric disorders were excluded from the study? Please, clarify this issue.
Table 3 is incomplete. Please, provide a complete version of the table.
Please, provide the definition of the abbreviations used in the Tables.
Discussion
The sample size is very small, even more in the follow-up survey. Please, include this fact as a limitation of the study.
Round 2
Reviewer 3 Report
The authors replied to reviewer's queries. I have no further comments